# Neuroprotective Effects of Noncanonical PAR1 Agonists on Cultured Neurons in Excitotoxicity

**DOI:** 10.3390/ijms25021221

**Published:** 2024-01-19

**Authors:** Irina Babkina, Irina Savinkova, Tatiana Molchanova, Maria Sidorova, Alexander Surin, Liubov Gorbacheva

**Affiliations:** 1Faculty of Medical Biology, Pirogov Russian National Research Medical University of the Ministry of Health of the Russian Federation, 117997 Moscow, Russia; irinababkina09@yandex.ru (I.B.); irenesavik@mail.ru (I.S.); 2Faculty of Biology, Lomonosov Moscow State University, 119991 Moscow, Russia; molchanovata94@gmail.com; 3Chazov National Medical Research Center for Cardiology, Ministry of Health of the Russian Federation, 121552 Moscow, Russia; mvs.peptide@gmail.com; 4Laboratory of Fundamental and Applied Problems of Pain, Institute of General Pathology and Pathophysiology, Russian Academy of Sciences, 119991 Moscow, Russia; surin_am@mail.ru

**Keywords:** activated protein C, neuroprotection, protease-activated receptors, neurons, glutamate, intracellular calcium

## Abstract

Serine proteases regulate cell functions through G protein-coupled protease-activated receptors (PARs). Cleavage of one peptide bond of the receptor amino terminus results in the formation of a new N-terminus (“tethered ligand”) that can specifically interact with the second extracellular loop of the PAR receptor and activate it. Activation of PAR1 by thrombin (canonical agonist) and activated protein C (APC, noncanonical agonist) was described as a biased agonism. Here, we have supposed that synthetic peptide analogs to the PAR1 tethered ligand liberated by APC could have neuroprotective effects like APC. To verify this hypothesis, a model of the ischemic brain impairment based on glutamate (Glu) excitotoxicity in primary neuronal cultures of neonatal rats has been used. It was shown that the nanopeptide NPNDKYEPF-NH2 (AP9) effectively reduced the neuronal death induced by Glu. The influence of AP9 on cell survival was comparable to that of APC. Both APC and AP9 reduced the dysregulation of intracellular calcium homeostasis in cultured neurons induced by excitotoxic Glu (100 µM) or NMDA (200 µM) concentrations. PAR1 agonist synthetic peptides might be noncanonical PAR1 agonists and a basis for novel neuroprotective drugs for disorders related to Glu excitotoxicity such as brain ischemia, trauma and some neurodegenerative diseases.

## 1. Introduction

The cerebral blood circulation decrease in ischemic stroke is one of the main pathological factors that leads to oxygen–glucose deprivation and glutamate excitotoxicity followed by neuronal cell death [1,2], inflammation [3] and BBB disruption that exacerbate brain injury [4]. 

A classic paradigm for G protein-coupled receptor (GPCR) activation was based on the understanding that receptor-bound agonists trigger or stabilize receptor-related changes until they reach an active conformation. In the last decade, it has been the dominant view that activation with different ligands may result in distinct active receptor conformations with unique divergent signaling profiles [5,6,7]. Recognition of these biased ligands resulted in a deeper understanding of mechanisms underlying biased agonism, improved assessment of ligand efficiency and advanced search and synthesis of novel ligands for clinical use. 

Acute ischemic stroke and brain injury are accompanied by the initiation of blood coagulation and the activation of hemostatic serine proteases, in particular, thrombin [8] and activated protein C (APC) [9]. Moreover, a special type of receptor—protease-activated receptors (PARs)—that mediate thrombin and APC-dependent regulation of cell functions can be involved in brain ischemia [10,11]. PARs are a family of highly conserved GPCRs activated by proteolytic cleavage. Currently, the role played by a biased agonism for APC and other hemostatic proteases acting on PARs has been extensively investigated and discussed [12,13,14]. Proteases cleave one peptide bond of the receptor amino terminus, which results in the formation of a new N-terminus (“tethered ligand”) capable of specific interaction with the second extracellular loop and activating the PAR [15]. Thrombin cleaves peptide bond Arg41–Ser42 in the extracellular N-terminal sequence LDPR41S42FLLRN of PAR1, thereby disclosing a new N-terminal peptide [15,16]. However, in contrast to thrombin, APC hydrolyzes peptide bond Arg46–Asn47 in the extracellular N-terminal sequence LDPRS FLLR46N47PNDKYEP of PAR1, disclosing a new N-terminal peptide NPNDKYEP—“tethered ligand” [12]—responsible for the cytoprotective activity of APC on endothelial cells and neurons [17]. Effects of APC can be mimicked by synthetic peptide analogs of the tethered ligands that are created after PAR1 cleavage at Arg46. By examining human endothelial cell line EA.hy926, it is found that 30 min of exposure with peptide TR47 resembling PAR1 residues 47–66 results in phosphorylation (inhibition) of glycogen synthase kinase 3β (GSK3β) at Ser9. TR47-induced GSK3β phosphorylation is inhibited by PAR1 antagonist SCH79797, suggesting that PAR1 is necessary for TR47-induced signaling [12]. However, cleavage of PAR1 by thrombin at Arg41 induces phosphorylation of extracellular regulated kinase (ERK1/2). The canonical PAR1 agonist peptide TRAP liberated by thrombin (TFLLRNPNDK) rapidly induces ERK1/2 phosphorylation, whereas TR47 does not [12]. Thrombin and APC, interacting with the same PAR1 receptor, exert multidirectional effects during excitotoxicity and inflammation [18,19]. Thrombin increases the expression of proinflammatory and proapoptotic factors together with the procoagulant effect [19]. APC is a neuroprotector in stressed neurons and hypoxic brain endothelium [18,20]. Recently, we have reported on the anti-apoptotic effects of APC on hippocampal neurons in glutamate excitotoxicity [21]. APC (10 nM) was shown to prevent the development of apoptosis induced in cortical neurons by NMDA and staurosporine [22]. By activating PAR1, APC controls the gene expression of proinflammatory and proapoptotic factors, stabilizes endothelial cells and neurons and protects them from death [18,19,20].

It has been shown recently that the multidirectional effects of thrombin and APC may be due to the biased agonism under the action of thrombin and APC on the PAR1 in endothelial cells [12,23]. Cleavage of PAR1 by APC at Arg46 results in GSK3β phosphorylation at Ser9, whereas PAR1 hydrolyzation by thrombin at Arg41 results in phosphorylation of ERK1/2 [24]. Activation of PAR1 by thrombin can lead to concomitant activation of the Gi/o, Gq and G12/13 families of G proteins, leading to various signaling pathways that ultimately result in transient endothelial barrier disruption. This transient endothelial barrier disruption is an important physiological response promoted by the activation of PAR1 receptors that couple to multiple heterotrimeric G protein subtypes including Gq/11 and G12/13 proteins. Biased agonism leads to the induction of distinct signaling mechanisms, such as the activation of PI3K, Akt and Rac1 by APC, which results in neuroprotection [25]. APC- or TR47-induced activation of PAR1 stabilizes different conformers of PAR1 that preferentially interact with β-arrestin-2. It is known that β-arrestins play a key role in desensitizing PARs [26]. By activating caveolar PAR1 bound to β-arrestins, APC results in the dissociation of receptor and adaptor proteins [23]. β-Arrestin is required for activation of small GTPase Rac1, which is a key mechanism in the accomplishment of the APC-related cytoprotective effect. PAR1-dependent β-arrestin-2 signaling via dishevelled-2 (Dvl-2) involves the activation of downstream signaling pathways such as PI3K/Akt and Rac1 and inhibits NF-kB, which promotes cell survival and enhances barrier integrity [27]. Overall, the interaction between PAR1, EPCR and S1P1 signaling pathways plays a critical role in mediating the cytoprotective effects of APC and thrombin in various physiological and pathological conditions [28]. Examining functions of proteases and PAR1 peptide agonists as well as analogs of tethered ligands liberated by thrombin or APC will increase understanding of the essence of biased agonism and outline ways for controlling PAR1-induced signaling mechanisms and cellular responses in inflammation, proliferation, tissue regeneration and neurotoxicity.

We have supposed that new synthetic nanopeptide NPNDKYEPF-NH2 (AP9) analogs of the PAR1 tethered ligand liberated by APC may also have a neuroprotective effect similar to that of APC. We used the model of glutamate excitotoxicity to simulate ischemic brain damage. In the case of brain ischemia, glutamate (Glu) is released massively into the intercellular space, which leads to the hyperactivation of pre- and postsynaptic glutamate receptors. The subsequent increase in intracellular free Ca^2+^ concentration ([Ca^2+^]i) may result in mitochondrial dysfunction, generation of reactive oxygen species and activation of proteases, phosphatases and endonucleases [29]. Massive influx of Ca^2+^ into the nerve cells through the channels of ionotropic Glu receptors disturbs the intracellular Ca^2+^ homeostasis and triggers the cascade of intracellular reactions, which end up with rapid or delayed cell death via the mechanisms of necrosis or apoptosis [30,31]. Undoubtedly, ischemia is a complex process that includes multiple factors in addition to excitotoxicity. However, any ischemic brain impairment is followed by excitotoxicity that leads to brain damage and cell death [32,33]. Comprehensive assessment of all pathological factors following brain ischemia was not the aim of this particular study, as in our manuscript, we tried to analyze and determine the mechanism of cytoprotective action by AP9 at the cellular level against the background of isolated action of ischemic damage factors caused by glutamate-induced excitotoxicity. The prolonged exposure of primary neuronal cultures to excitotoxic Glu concentrations results, eventually, in a secondary rise of [Ca^2+^]i (delayed calcium dysregulation, DCD) followed by a high [Ca^2+^]i plateau [32]. The DCD is accompanied by synchronous profound mitochondrial depolarization (MD) and a secondary decrease in mitochondrial NADH and pH [34,35]. These injurious processes include Ca^2+^ overload of mitochondria, reactive oxygen and nitrogen species formation, activation of caspases and release of apoptosis-inducing factor [36,37]. Neurons having DCD die in a few hours by necrosis or apoptosis [38,39].

The search for agents that help the cells to resist excitotoxicity and studies of the mechanisms triggered by these agents are necessary for the optimization of therapy for acute neurological diseases. Although the protective effects of APC against Glu excitotoxicity are well documented [18,20,21,22], the data about the function of hemostatic proteinases and their receptors in the central nervous system are still contradictory [19,40,41,42,43,44,45,46,47]. The goal of the present work was to identify the possible neuroprotective properties of a new synthetic peptide AP9 and to compare them with the effects of APC in primary neuronal cultures subjected to glutamate excitotoxicity.

## 2. Results

### 2.1. The Toxic Effects of Glutamate and NMDA on the Cultured Neurons

In this study, we used the widely used model of glutamate-induced neurotoxicity [21,48,49]. The exposure of cultured rat cortical neurons to glutamate (Glu, 100 µM, 40 min) led to a significant decrease in cell survival. With MTT assay, a 40% decrease in cell survival 24 h after treatment with Glu was shown (Figure 1). Glutamate excitotoxicity is mostly due to calcium overload of the cytoplasm and mitochondrial dysfunction caused by calcium and sodium influx through the NMDA subtype of glutamate receptors [50,51,52]. Blockage of NMDARs by MK-801 abolished glutamate-induced cell death (Figure 1). These results confirm that glutamate cell death in neurons is mediated by NMDARs [53].

### 2.2. The Effects of APC and AP9 on Survival of Cultured Neurons in Glutamate Excitotoxicity

In the present work, preincubation of cultured cortical neurons in the presence of 10 nM APC resulted in a more than 1,4-fold increase in the number of living cells (Figure 2) in glutamate excitotoxicity. These data are in accordance with previous observations that APC has neuroprotective properties in staurosporine- or NMDA-induced toxicity [18,20].

In order to estimate the neuroprotective effects of a new nanopeptide (AP9) liberated by APC from the N-terminal sequences of PAR1 exodomain, we examined the survival of neurons during glutamate-induced excitotoxicity in the presence of different concentrations of AP9. It was found that AP9 at concentrations of 20 and 200 µM significantly reduces cell death in Glu-induced toxicity (Figure 2). Thus, the protective effects of AP9 were similar to APC’s impact on neurons. It has been shown that the number of living neurons in the presence of peptides does not significantly differ from their number in the presence of APC (Figure 2). These results confirm that AP9 nanopeptide at low concentrations (2, 200 μM) has similar neuroprotective effects as APC, consistent with previous reports [54].

### 2.3. PAR1 Is Required for Protective Effects of AP9 and APC in Glutamate-Induced Excitotoxicity

Previously, it was shown that PAR1 or the PAR/EPCR complex is required for the realization of the cytoprotective effects of APC [55,56,57]. Taking into consideration the similarities in the neuroprotective effects between APC and AP9, which is an analog to tethered ligand liberated by APC, we suggested that the AP9 might also realize its protective effects through the same type of receptor as APC—PAR1. In the next series of experiments, the estimation of cell survival was made using the MTT test during Glu excitotoxicity in the presence of AP9 and/or in the presence of a PAR1 inhibitor—SCH79797 (Figure 3). Preincubation with APC (10 nM) or AP9 (20 μM) significantly increased hippocampal neuron survival in Glu excitotoxicity. The blockage of PAR1 by SCH79797 abolished the neuroprotective effect of both APC and AP9, demonstrating the necessity of PAR1 for the protective effects of these substances.

### 2.4. The Effects of APC and AP9 on the Intracellular Free Ca^2+^ Concentration ([Ca^2+^]i) Dysregulated by Glu and NMDA

Glu (100 µM, 15 min)-induced delayed calcium dysregulation (DCD) was followed by a sustained 2 sec plateau. [Ca^2+^]i was altered in hippocampal neurons by NMDA (300 µM, 15 min) application similarly to the glutamate effects (Figure 4A). The pretreatment of the cells with PAR1 agonists—APC and AP9—significantly decreased the maximum response of [Ca^2+^]i to the toxic Glu concentration. Moreover, pretreatment of neurons with AP9 (20 μM) before the application of NMDA (300 μM) leads to the stabilization of calcium homeostasis that is expressed in a significant decrease in the area under the curve (Figure 4B). As the glutamate-induced calcium flow triggers calcium overload and neurotoxicity [58], the reduction in initial calcium influx may illustrate the protective effects of APC. AP9 was also shown to restore the Glu- and NMDA-induced [Ca^2+^]i dysregulation; however, the exact mechanisms of its regulatory effects are under debate.

## 3. Discussion

In the present work, we have studied the protective effects of PAR1-biased activation by APC and a new synthetic peptide AP9 in glutamate-induced neurotoxicity. Previously, we have shown the protective effect of APC on the survival of hippocampal neurons exposed to toxic doses of glutamate [21]. Here, we demonstrated that APC prevents Glu-excitotoxicity via PAR1 in primary hippocampal neurons (Figure 3). Recently, it was shown that thrombin and APC have distinctly different properties because each is able to stabilize a different subset of the dynamic conformational ensembles of PAR1 [12]. While thrombin and thrombin receptor activating peptide (TRAP) promote PAR1 signaling via different G-proteins, APC-induced activation promotes signaling via β-arrestin and dishevelled-2 [12,23]. Compared to thrombin, APC allosterically modulates PAR1 [14,56]. APC-cleaved PAR1 is localized in caveolae, plasma membrane microdomains, lipid rafts enriched in cholesterol and caveolin-1. Herein, APC-activated PAR1 is colocalized on the endothelial membrane with EPCR bound to caveolin-1 and is necessary for cytoprotective functions accomplished by APC [59,60]. To demonstrate that PAR1-dependent signaling by APC involves a novel cleavage of the receptor’s N-terminal domain, differing from that of thrombin, we used a new synthetic peptide analog of the tethered ligand liberated by APC—AP9 (NPNDKYEPF-amide). We studied the effect of the AP9 on neuronal survival under the influence of glutamate in comparison with APC and we measured changes of [Ca^2+^]i under the influences of glutamate and NMDA as an indicator of high neuronal sensibility to the used cytotoxic glutamate concentrations [50,61,62,63].

In our work, we proved that the presence of APC and AP9 protects neurons and restores the basal level of calcium, significantly increased during glutamate-induced excitotoxicity (Figure 4). PAR1 is required for neuroprotective action of APC and AP9 (Figure 3). Earlier, we showed that APC prevents neuronal death by decreasing the translocation of NF-kBp65 into the nucleus and abolishing the increase in proapoptotic proteins in excitotoxicity [55]. Here, we show for the first time in hippocampal and cortical neurons that AP9 demonstrates a protective effect similar to APC. The novel peptide agonist of PAR1 increases the survival of neurons in culture after glutamate-induced toxicity (Figure 2 and Figure 3). Thus, our data support the hypothesis that APC’s cleavage of PAR1 of hippocampal and cortical neurons occurs at Arg46 and agrees with another recent report which advances the paradigm that APC’s PAR1-dependent protective actions are based on Arg46 cleavage. These studies showed that the TR47 peptide representing the sequence of the novel N-terminus that is generated by cleavage at Arg46 exerts remarkable cytoprotective activities on endothelial cells and HEK [64]. This peptide promotes cytoprotective signaling via β-arrestins, especially β-arrestin-2, and dishevelled-2 (Dvl-2) scaffold [12,56]. The advantage of our AP9 peptide in comparison to other synthetic analogs is its small size, which is preferable in terms of synthesis. Our present results corroborate well and extend another report [65] that demonstrates the β-arrestin-2-dependent protective properties of a PAR1 agonist peptide, AP9, in vivo in a mouse model of photothrombosis-induced brain ischemia. 

Excessive entry of Ca^2+^ through the NMDA receptor is thought to be the major cause of glutamate toxicity in brain neurons [66]. Here, we have shown that NMDARs are involved in Glu toxicity in neuronal cell culture, consistent with previous reports [51,67]. Moreover, we demonstrated for the first time that the protective effect of AP9 peptide is realized through the stabilization of Glu- and NMDA-induced dysregulation of intracellular calcium. At the same time, the effects of AP9 and APC were codirected and comparable (Figure 4).

Our interest in NMDAR was induced by the data of previous studies that pointed to the possibility of PAR1-mediated NMDAR potentiation by thrombin [68]. Thrombin and other serine proteases can enter into the brain parenchyma during intracerebral hemorrhage or the extravasation of plasma proteins during blood–brain barrier breakdown, which may exacerbate glutamate-mediated cell death and possibly participate in post-traumatic ischemic injuries. For years, PAR1 has been regarded as a positive modulator of NMDAR potentiation and an important mechanism for seizure initiation and subsequent neurodegeneration. 

We have suggested the possibility of similar PAR1-mediated effects of APC and AP9 on NMDAR that, in contrast to thrombin, lead to the stabilization of intracellular calcium homeostasis and neuroprotection (Figure 5). One of the neurodegenerative mechanisms is the pathological, persistent elevation of intracellular calcium. Previously, we have shown in the model of glutamate excitotoxicity that the protective effects of APC are associated with the determination of calcium signaling and the transcriptional factor NF-kB [55]. In this work, we have tried to analyze the effect of AP9 peptide on cytosolic calcium, which is one of the factors leading to neuronal death. We demonstrated that under the influence of AP9, neuronal death decreases and the level of intracellular calcium stabilizes, so on the basis of this, we can conclude that the protective mechanism of AP9 is calcium-dependent. Though the exact mechanism underlying the neuroprotective effects of AP9 has not been discovered yet, it might function similarly to TR47, which realizes its effects via β-arrestins, especially β-arrestin-2, and dishevelled-2 (Dvl-2) scaffold [12,56]. The present data revealed that biased agonists of PAR1 may be new candidates for neuroprotective drugs, and PAR1-dependent signaling has a wider variety of pathways than previously known. Thus, understanding the molecular mechanisms of the protective action and the peptide structure of noncanonical PAR1 agonists may facilitate the development of new therapeutic pathways for neurodegenerative brain damage.

## 4. Materials and Methods

### 4.1. Reagents

Human APC, NaCl, KCl, CaCl_2_, MgCl_2_, KH_2_PO_4_, HEPES, glucose, glutamate, NMDA, Ara C and PAR1 inhibitor SCH79797 were from Sigma-Aldrich (St. Louis, MO, USA). Neurobasal medium A (NBM), Supplement B27 and L-GlutaMax were from Gibco (Thermo Fisher Scientific, 168 Third Avenue, Waltham, MA, USA). AP9 (NPNDKYEPF-amide) was synthesized at the Laboratory of peptide synthesis of the Russian Cardiology Research and Production Complex by the standard technology of solid-phase peptide synthesis using the Fmoc (9-fluorenemethoxycarbonyl) strategy. The structure and homogeneity of the peptide were confirmed by H-NMR spectroscopy and analytical HPLC.

### 4.2. Preparation of Cell Cultures

Experiments with animals were performed in accordance with the ethical principles and regulatory documents recommended by the European Convention on the Protection of Vertebrate Animals used for experiments [69], as well as in accordance with the “Good Laboratory Rules practice”, approved by order of the Ministry of Health of the Russian Federation No. 199n of 04/01/2016. 

Primary cultures of rat brain hippocampal and cortical neurons were prepared from 1–2-day-old Wistar rats, as described earlier [35,49]. Briefly, the rats were anesthetized and decapitated, and the hippocampus or cortex was removed and separated from the meninges. The extracted tissues were washed in a Ca^2+^- and Mg^2+^-free Hanks solution, crushed and placed in a papain solution for 15 min at 36 °C, washed with standard Hanks solution and Minimal Essential Medium (MEM) and dispersed in fresh MEM. A homogeneous suspension was precipitated in a centrifuge at 200 *g* for 5 min. The precipitated cells were resuspended to a concentration of 10^6^ cells/mL in neurobasal medium (NBM), supplemented with 2% B-27, 1% GlutaMAX and 1% penicillin/streptomycin (NBM+). The suspension (200 µL) was transferred onto coverslips attached to the wells of 35 mm plastic glass-bottom Petri dishes (El Segundo, CA, USA) or into the wells (400 µL/well) of 24-well plastic plates Costar^®^ Multiple Well Cell Culture Plates (5310 W. Camelback Rd., Glendale, AZ, USA). After 1 h, 1.5 mL of NBM+ was added to each dish. The dishes and plates were precoated with 10 mg/mL of polyethyleneimine. The cells were kept in an incubator at 37 °C, 95% air, 5% CO_2_ and a relative humidity of 100% until use on days 9–10 in vitro (DIV). Cytosine arabinoside (AraC, 5 µM) was added to the medium for two or three days to prevent the proliferation of glial cells. Every three days, the cells were fed by replacing 1/3 of the old medium with a fresh NBM+.

### 4.3. Cytotoxicity Assays

A biochemical MTT assay and morphological method employing fluorescence vital dyes were used to estimate cell viability. A colorimetric MTT assay is based on the reduction of the yellow 3-(4,5-dimethylthiazol-2-yl)-2,5-diphenyl-tetrazolium bromide (MTT), mostly by mitochondria of living cells, to dark-blue formazan [70,71].

Excitotoxicity was reached by the substitution of NBM+ with buffered saline solutions (HBSS) containing glutamate (100 µM) or N-methyl-D-aspartate (NMDA, 200 µM) for 40 min at 37 °C. Control HBSS included (mM) NaCl, 145; KCl, 5; CaCl_2_, 1.8; MgCl_2_, 1.0 HEPES, 20; glucose, 5 (pH 7.4). Glycine (10 µM) was added and MgCl_2_ was omitted in HBSS containing Glu or NMDA. APC (10 nM) or AP9 (2, 20, 200 µM) were added alone to HBSS or 15 min prior to Glu and NMDA. NBM+ was aspirated from the cells seeded in a 24-well plate, and then HBSS buffers containing Glu, NMDA, APC and AP9 in appropriate combinations were added to the corresponding wells. Next, cells were washed with saline, NBM+ was returned to the wells and the cells were put back in the CO_2_ incubator for 24 h.

After 24 h, NBM+ was removed from all 24 wells of the plate and 0.5 mL of MTT in HBSS (4 mg/mL) was added to each well. In 30 min, the MTT-containing buffer was aspirated, and formazan was dissolved in 300 mL of DMSO. The light absorbance of the formazan solution was measured at 550 nm (A550) using a plate reader ClarioStar (BMG LABTECH, Allmendgrün 8, Ortenberg, Germany). Absorbance at 650 nm was subtracted from A550 to compensate for light absorbance and scattering by the bottom of the plastic plate. The optical density of the control group and cell-free wells were considered as 100 and 0% survival, respectively.

### 4.4. Fluorescence Imaging Analysis

For the morphological determination of neuronal death, cells were incubated concomitantly for 30 min with permeant fluorescent dyes, DNA intercalating Hoechst 33342 (10 µg/mL; excitation, 360 nm; emission, 460 nm; Molecular Probes, Eugene, OR, USA) and the DNA/RNA-specific dye SYTO-13 (1 µM; excitation, 488 nm; emission, 520 nm; Molecular Probes, Thermo Fisher Scientific, 168 Third Avenue, Waltham, MA, USA). Nuclear staining was visualized using fluorescence microscopy (Axiovert 200; Zeiss, Jena, Germany). The cells with pycnotic and incorrectly shaped nuclei were considered apoptotic. Normal nuclei (living cells) exhibited loose chromatin colored green by SYTO-13 and blue by Hoechst 33342, and their number was expressed as a percent of the total cell number, with the number of living cells in the control group taken as 100% [72]. Five randomly chosen fields were analyzed from each glass.

### 4.5. Intracellular Free Ca^2+^ ([Ca^2+^]i) Measurements

The experimental setup included a ZEISS LSM 700 confocal microscopy. To measure [Ca^2+^]i, the cells were loaded with high-affinity Ca^2+^ indicator Fluo-4 in the form of the acetoxymethyl (AM) ester (1–2 μM Fluo-4, 40 min, 37 °C). Fluo-4 fluorescence was excited at 488 nm and monitored at 505–535 nm. All measurements were carried out at 27–29 °C in HBSS. Changes of [Ca^2+^]i induced by Glu or NMDA were measured in Mg^2+^-free glycine-containing buffers, as mentioned above. Glu or NMDA were washed out by a nominally calcium-free solution containing 0.1 mM EGTA instead of CaCl_2_ and 2 mM MgCl_2_. Replacement of the solutions was performed by quickly (<13 s, 2 × 200 mkl) sucking the previous buffer out and adding the new one into the dish with the cells. To compare relative changes in [Ca^2+^]i induced by Glu or NMDA in experiments performed on different days, we calibrated the fluorescence signals of Fluo-4. To this end, during the final part of the experiments, a Ca^2+^ ionophore ionomycin (2 μM) was applied in the presence of 5 mM Ca^2+^ to saturate the indicator with Ca^2+^ and measure its maximum signals.

### 4.6. Data Processing

The data from 4–6 independent experiments were analyzed using GraphPad Prism 8 (GraphPad, San Diego, CA, USA). The data were processed in paired samples using Student’s *t*-test. Cytosolic calcium levels were compared using two-way ANOVA (Dunnett’s test). The differences were considered significant at *p* < 0.05; n was the number of independent experiments. The results are presented as the mean with the standard error of the mean.

## 5. Conclusions

For the first time, the neuroprotective effect of the new nanopeptide analog to the “tethered ligand” liberated by APC from PAR1 was demonstrated in this research under excitotoxicity. These findings lead to the conclusion that the protective effects of this peptide are comparable with the protective effects of APC under glutamate-induced excitotoxicity. We have previously demonstrated the protective effect of the peptide AP9 in a model of photothrombosis-induced ischemia in vivo [65], and here we have attempted to study the possible mechanism of this protective effect. To apply this synthetic peptide to clinical use, a long series of studies is needed to determine the mechanism of its action and identify the range of therapeutic concentrations. However, at this moment, it is too early to provide an assessment of the clinical use of this peptide, despite the undeniable fact that PAR1 agonist synthetic peptides are a promising basis for novel neuroprotective drugs for disorders related to glutamate excitotoxicity. The application of low-molecular-mass compounds in contrast to large proteins decreases the possibility of immunological reactions and increases the success of the solution to the problem of “addressed” drug delivery. The study of the mechanisms of PAR1 agonist peptide action, as well as the development of new modifications of noncanonical PAR1 agonists with high neuroprotective activity, may be an important and relevant trend in the search for novel neuroprotective agents for treating neurodegenerative diseases and stroke.

## Figures and Tables

**Figure 1 ijms-25-01221-f001:**
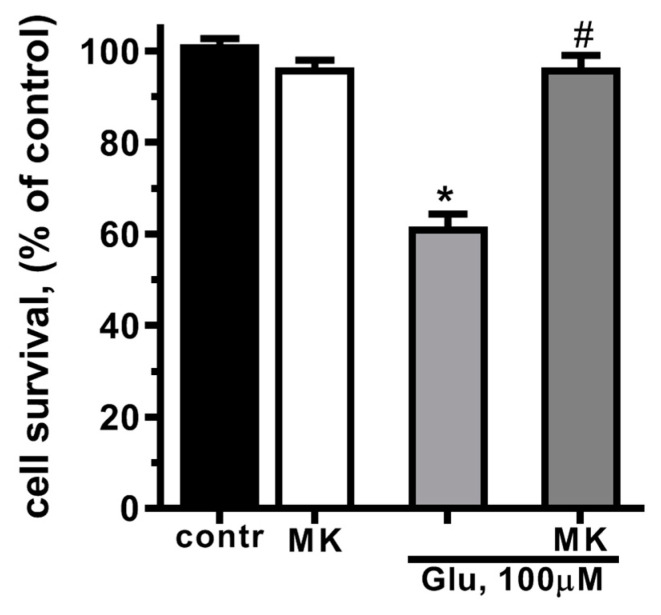
Activation of NMDA glutamate receptors decreases cultured cortical neuron survival. Cells were cultured in NBM+ during days 9–10. Excitotoxicity was reached by substitution of NBM+ for buffered saline solutions (HBSS) containing glutamate (100 µM) for 40 min at 37 °C. NMDAR blocker MK-801 (10 µM) was added alone to HBSS or 15 min prior to Glu. After 24 h, NBM+ was removed and cell viability was detected with the MTT assay kit as described in Section 4.3. Values are expressed as the mean ± S.D. of triplicate cultures. *—*p* < 0.05 compared to control group, #—*p* < 0.05 compared to group with glutamate (contr—with control HBSS containing 145 mM NaCl; 5 mM KCl; 1.8 mM CaCl_2_; 1.0 mM MgCl_2_; 20 mM HEPES; 5 mM glucose (pH 7.4), MK—with MK-801).

**Figure 2 ijms-25-01221-f002:**
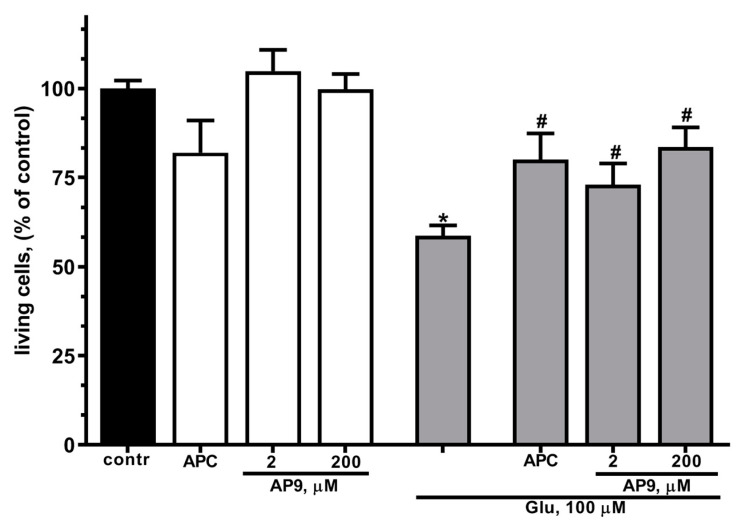
AP9 increases survival of cultured cortical neurons under excitotoxicity more efficiently than APC. Cells were cultured in NBM+ during days 9–10. Excitotoxicity was reached by substitution of NBM+ for buffered saline solutions (HBSS) containing glutamate (100 µM) for 40 min at 37 °C. APC (10 nM) or AP9 (2 or 200 µM) were added alone to HBSS or 15 min prior to Glu. After 24 h, NBM+ was removed and morphological estimation of cell viability was performed using fluorescent dyes Syto-13 and Hoechst 33342 (see Section 4.4). Values are expressed as the mean ± S.D. of triplicate cultures. *—*p* < 0.05 compared to control group, #—*p* < 0.05 compared to group with glutamate (contr—with control HBSS containing 145 mM NaCl; 5 mM KCl; 1.8 mM CaCl_2_; 1.0 mM MgCl_2_; 20 mM HEPES; 5 mM glucose (pH 7.4)).

**Figure 3 ijms-25-01221-f003:**
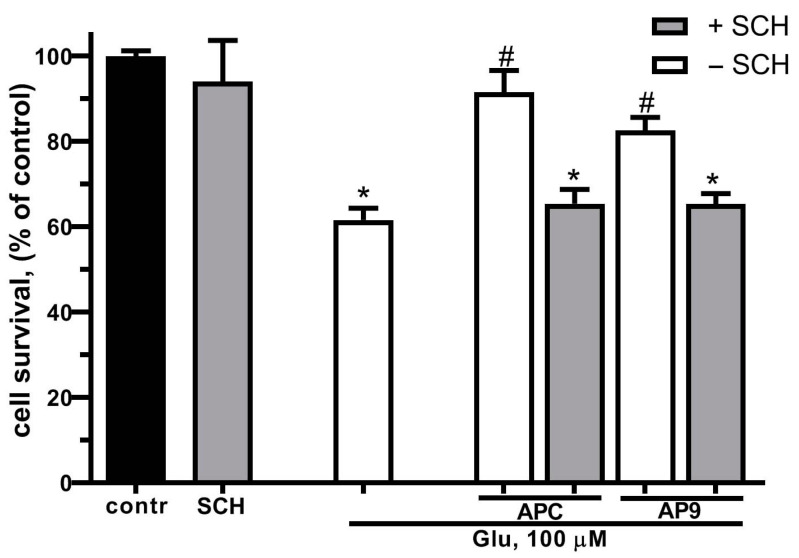
Effect of PAR1 blockage on hippocampal neuronal survival in glutamate toxicity and APC/AP9 treatment. Cells were cultured in NBM+ during days 9–10. Excitotoxicity was reached by substitution of NBM+ with buffered saline solutions (HBSS) containing glutamate (100 µM) for 40 min at 37 °C. PAR1 blocker SCH79797 (50 nM) was added to HBSS 30 min prior to Glu; APC (10 nM) or AP9 (20 µM) were added to HBSS 15 min prior to Glu. After 24 h, NBM+ was removed and cell viability was detected with the MTT assay kit as described in Section 4.3. Values are expressed as the mean ± S.D. of triplicate cultures. *—*p* < 0.05 compared to control group, #—*p* < 0.05 compared to group with glutamate (contr—with control HBSS containing 145 mM NaCl; 5 mM KCl; 1.8 mM CaCl_2_; 1.0 mM MgCl_2_; 20 mM HEPES; 5 mM glucose (pH 7.4), SCH—SCH79797).

**Figure 4 ijms-25-01221-f004:**
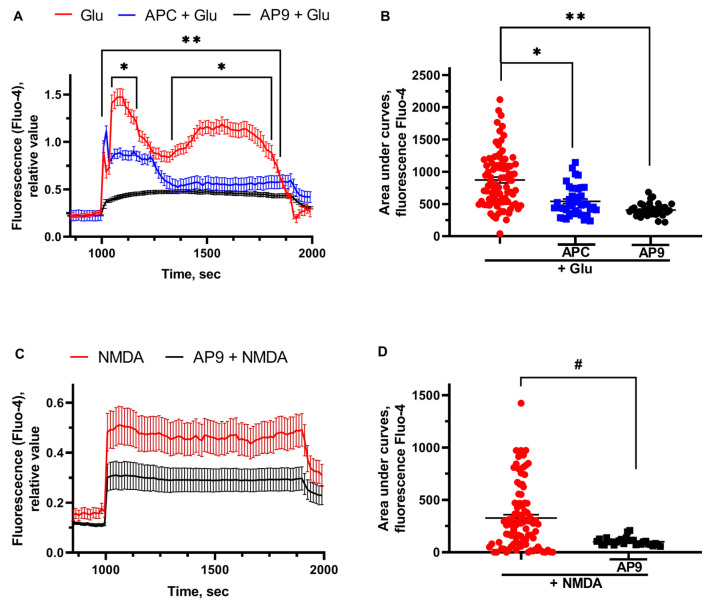
APC and AP9 restore the Glu- and NMDA-induced [Ca^2+^]i dysregulation in hippocampal neurons. (**A**) Glutamate-induced [Ca^2+^]i alterations in cultured neurons and (**B**) changes of the areas under curves measured with Ca-dependent fluorescence Fluo-4 (effects of glutamate and PAR1 agonists), (**C**) NMDA-induced [Ca^2+^]i alterations in cultured neurons and (**D**) changes of the areas under curves measured with Ca-dependent fluorescence Fluo-4 (effects of NMDA and PAR1 agonists). For [Ca^2+^]i measurement, the cells were loaded with high-affinity Ca^2+^ indicator Fluo-4 in the form of the acetoxymethyl (AM) ester (1–2 μM Fluo-4, 40 min, 37 °C). Fluo-4 fluorescence was excited at 488 nm and monitored at 505–535 nm. All measurements were carried out at 27–29 °C in HBSS. Glu or NMDA were washed out by a nominally calcium-free solution containing 0.1 mM EGTA instead of CaCl_2_ and 2 mM MgCl_2_. For the measurement of the maximal signal at the final part of the experiments, a Ca^2+^ ionophore ionomycin (2 μM) was applied in the presence of 5 mM Ca^2+^ to saturate the indicator with Ca^2+^. *—*p* < 0.05 APC pretreatment compared to glutamate, **—*p* < 0.05 AP9 pretreatment compared to glutamate, #—*p* < 0.05 AP9 pretreatment compared to NMDA.

**Figure 5 ijms-25-01221-f005:**
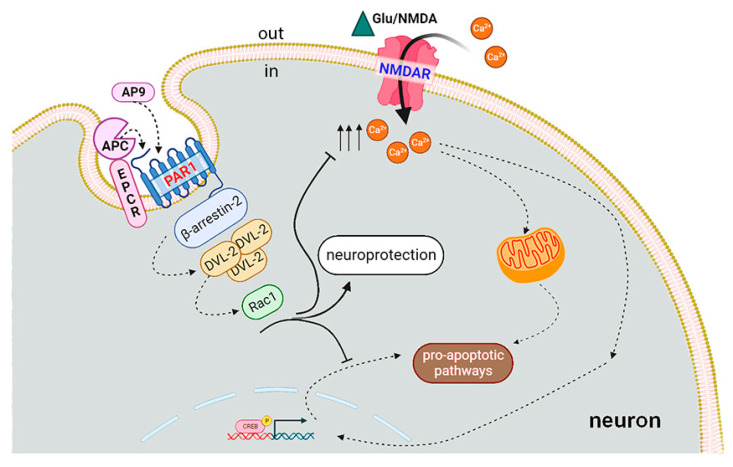
Biased signaling at PAR1 induced by APC and nanopeptide NPNDKYEPF-NH2 (AP9) analogs of the PAR1 tethered ligand liberated by APC. Though the exact mechanism underlying the neuroprotective effects of AP9 has not been discovered yet, it might function similarly to APC, promoting cytoprotective signaling via β-arrestin-2 and dishevelled-2 (Dvl-2) scaffold. In the present study, it was shown that the protective effects of APC and AP9 peptide are realized through stabilization of Glu- and NMDA-induced dysregulation of intracellular calcium concentration.

## Data Availability

Data are contained within the article.

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
