# Peer review of "Neuroprotective Effects of Noncanonical PAR1 Agonists on Cultured Neurons in Excitotoxicity"

_ijms, 2024, doi:10.3390/ijms25021221_

Round 1

Reviewer 1 Report

Comments and Suggestions for Authors

Authors aim to investigate potential neuroprotective properties of newly synthesized nanopeptide, analog to PAR1 using primary cortical neurons cultures. The model itself is adequate to the hypothesis testing neuroprotective properties of AP9 against excitotoxicity. However, submission as whole remains standard screening test seen in testing bio-properties of new compounds. Results presented clearly demonstrate that AP9 acts through limitation  of Ca influx into excitotoxicity- activated neurons. Thereby, it increases neuronal survival/mitochondrial oxidations in these conditions, and mimics APC -  natural inhibitor of Ca channel.

However, there are some setbacks, which should be corrected.

There is no experimental data concerning hippocampal neurons. Thus, all these information should be omitted(lines 251,254 and others).

MTT test reflects both cell number and their metabolic activity. Therefore, some direct cell counts should be performed to check whether alterations of MTT reduction result from: cell death, mitochondrial suppression, or both. Direct morphological analysis is also advisable.

…..cultural….. (line 264).

The sentences line 263-266 should be changed to underline that AP9 effects are mediated by AP9, what is new finding. Some newer Neuro-excitoxicity reviews are available. One should be added to ref. 48.

Graph demonstrating AP9 mechanism ought to be added.

Submission needs improvements listed above prior publication.

Author Response

Reviewer 1

The authors are grateful to the reviewer for the careful and insightful reading of the manuscript and helpful comments.

Reviewer. Authors aim to investigate potential neuroprotective properties of newly synthesized nanopeptide, analog to PAR1 using primary cortical neurons cultures. The model itself is adequate to the hypothesis testing neuroprotective properties of AP9 against excitotoxicity. However, submission as whole remains standard screening test seen in testing bio-properties of new compounds. Results presented clearly demonstrate that AP9 acts through limitation of Ca influx into excitotoxicity- activated neurons. Thereby, it increases neuronal survival/mitochondrial oxidations in these conditions, and mimics APC -  natural inhibitor of Ca channel.

However, there are some setbacks, which should be corrected.

Reviewer: There is no experimental data concerning hippocampal neurons. Thus, all these information should be omitted (lines 251,254 and others).

Author response: Thank you for the valuable suggestions and comments. We recognize our omission that we did not clearly identify the cell types in the text of the article. Some experiments were conducted on both hippocampal and cortical neurons (Fig. 3 and 4). That is the reason why in the discussion we declare the presence of APC and AP9 protective effects in both cell cultures. We have now made the appropriate changes to the text.  

Reviewer: MTT test reflects both cell number and their metabolic activity. Therefore, some direct cell counts should be performed to check whether alterations of MTT reduction result from: cell death, mitochondrial suppression, or both. Direct morphological analysis is also advisable.

Author response:  We are very grateful for your comment as we have found an error in the caption of Figure 2. In this case,  morphological estimation of cell viability was performed using fluorescent vital dyes Syto-13 (1 µM; excitation, 488 nm; emission, 520 nm) for living cells and Hoechst 33342 (10 µg/ml; excitation, 360 nm; emission, 460 nm) to visualize nuclei of living and apoptotic cells. According to our previous studies on neurons in the model of excitotoxicity, the MTT data and the results of morphological evaluation were comparable. (Gorbacheva LR, Storozhevykh TP, Pinelis VG, Davydova ON, Ishiwata S, Strukova SM. Activated protein C via PAR1 receptor regulates survival of neurons under conditions of glutamate excitotoxicity. Biochemistry (Mosc). 2008 Jun;73(6):717-24. doi: 10.1134/s0006297908060138. PMID: 18620539).Also, previous studies by Surin A.M. have demonstrated that the reduction of MTT to formazan in cultured neurons occurs predominantly in mitochondria and, probably, with participation of complex I in this reaction (Surin AM, Sharipov RR, Krasil'nikova IA, Boyarkin DP, Lisina OY, Gorbacheva LR, Avetisyan AV, Pinelis VG. Disruption of Functional Activity of Mitochondria during MTT Assay of Viability of Cultured Neurons. Biochemistry (Mosc). 2017 Jun;82(6):737-749. doi: 10.1134/S0006297917060104. PMID: 28601083)

Reviewer: …..cultural….. (line 264).

Author response: We have corrected the text: “…cultural neurons…” has been removed and “.. neuronal cell culture.. was added”, (see line 264).

Reviewer: The sentences line 263-266 should be changed to underline that AP9 effects are mediated by AP9, what is new finding. Some newer Neuro-excitoxicity reviews are available. One should be added to ref. 48.

Author response: We added a new reference (Shen Z, Xiang M, Chen C, Ding F, Wang Y, Shang C, Xin L, Zhang Y, Cui X. Glutamate excitotoxicity: Potential therapeutic target for ischemic stroke. Biomed Pharmacother. 2022 Jul;151:113125. doi: 10.1016/j.biopha.2022.113125. Epub 2022 May 24. PMID: 35609367). We have corrected the text: “…Moreover, we have demonstrated the significant impact of APC and AP9 on Glu- and NMDA-induced dysregulation of calcium hemostasis (Figure 4). …” has removed and added “..Moreover, we demonstrated for the first time that the protective effect of AP9 peptide is realized through stabilization of Glu- and NMDA-induced dysregulation of intracellular calcium. At the same time, the effects of AP9 and APC were co-directed and comparable (Fig. 4)…”

Reviewer: Graph demonstrating AP9 mechanism ought to be added.

Author response: The graph that demonstrate the possible AP9 mechanisms is added (Figure 5).

Reviewer 2 Report

Comments and Suggestions for Authors

1.    The study only used a model of ischemic brain impairment based on glutamate excitotoxicity in primary neuronal cultures of neonatal rats, which may not fully represent the complexity of brain ischemia in humans.

2.    The paper did not investigate the long-term effects of the synthetic peptide analogues on neuroprotection or their potential side effects.

3.    The study focused on the neuroprotective effects of noncanonical PAR1-agonists on cultured neurons, but did not explore their effects in vivo or in clinical settings.

4.    The paper did not compare the neuroprotective effects of the synthetic peptide analogues to other existing neuroprotective drugs or therapies.

5.    The study did not provide detailed information on the mechanisms underlying the neuroprotective effects of the synthetic peptide analogues.

6.    The paper did not discuss the potential limitations or challenges in translating the findings from in vitro experiments to clinical applications.

Author Response

The authors are grateful to the reviewer for carefully reading the manuscript and making comments.

  1. The study only used a model of ischemic brain impairment based on glutamate excitotoxicity in primary neuronal cultures of neonatal rats, which may not fully represent the complexity of brain ischemia in humans.

Author response: We totally agree with the reviewer that ischaemia is a complex process that includes multiple factors in addition to excitotoxicity. However, excitotoxicity is one of significant damage factors led to cell death at brain ischaemia (Diogo Neves, Ivan L. Salazar, Ramiro D. Almeida, Raquel M. Silva, Molecular mechanisms of ischemia and glutamate excitotoxicity, Life Sciences, V 328, 2023, 121814, https://doi.org/10.1016/j.lfs.2023.121814).  Our previously research demonstrated the protective effects of new PAR1-agonist during photo-induced ischaemia in in vivo model on mice (Galkov M, Kiseleva E, Gulyaev M, Sidorova M, Gorbacheva L. New PAR1 Agonist Peptide Demonstrates Protective Action in a Mouse Model of Photothrombosis-Induced Brain Ischemia. Front Neurosci. 2020 May 19;14:335. doi: 10.3389/fnins.2020.00335. PMID: 32547356; PMCID: PMC7273131). This study demonstrated the protective effects of new PAR1-agonist in vitro on neuronal cultures during excitotoxicity. The possible mechanism of the protective action of PAR1-agonist on is peptide-induced prevention of cytosolic calcium rise at glutamate-toxicity.

Corresponding clarifications, we have included to the text of the manuscript.

  1. The paper did not investigate the long-term effects of the synthetic peptide analogues on neuroprotection or their potential side effects.

Author response:

In this study, we used the glutamate-induced neurotoxicity model in its widely used modification  (Zhang LN, Wang Q, Xian XH, Qi J, Liu LZ, Li WB. Astrocytes enhance the tolerance of rat cortical neurons to glutamate excitotoxicity. Mol Med Rep. 2019 Mar;19(3):1521-1528. doi: 10.3892/mmr.2018.9799. Epub 2018 Dec 24. PMID: 30592287; PMCID: PMC6390011; Krasil'nikova I, Surin A, Sorokina E, Fisenko A, Boyarkin D, Balyasin M, Demchenko A, Pomytkin I, Pinelis V. Insulin Protects Cortical Neurons Against Glutamate Excitotoxicity. Front Neurosci. 2019 Sep 24;13:1027. doi: 10.3389/fnins.2019.01027. PMID: 31611766; PMCID: PMC6769071; Gorbacheva LR, Storozhevykh TP, Pinelis VG, Davydova ON, Ishiwata S, Strukova SM. Activated protein C via PAR1 receptor regulates survival of neurons under conditions of glutamate excitotoxicity. Biochemistry (Mosc). 2008 Jun;73(6):717-24. doi: 10.1134/s0006297908060138. PMID: 18620539). The duration of the study is determined by the specificity of the primary cultures. Other authors estimated effects of synthetic peptides composing PAR1 residues, like  TR47,  using time-course protocols comparable to ours (Mosnier LO, Sinha RK, Burnier L, Bouwens EA, Griffin JH. Biased agonism of protease-activated receptor 1 by activated protein C caused by noncanonical cleavage at Arg46. Blood. 2012 Dec 20;120(26):5237-46. doi: 10.1182/blood-2012-08-452169. Epub 2012 Nov 13. PMID: 23149848; PMCID: PMC3537315). The advantage of our AP9 peptide is its small size in comparison to other synthetic analogues, which is preferable in terms of synthesis.

Corresponding clarifications, we have included to the text of the manuscript.

  1. The study focused on the neuroprotective effects of noncanonical PAR1-agonists on cultured neurons, but did not explore their effects in vivo or in clinical settings.

Author response: We have previously demonstrated the protective effect of the peptide in a model of photothrombosis-induced ischaemia in vivo (Galkov M, Kiseleva E, Gulyaev M, Sidorova M, Gorbacheva L. New PAR1 Agonist Peptide Demonstrates Protective Action in a Mouse Model of Photothrombosis-Induced Brain Ischemia. Front Neurosci. 2020 May 19;14:335. doi: 10.3389/fnins.2020.00335. PMID: 32547356; PMCID: PMC7273131), and here we have attempted to evaluate the possible mechanism of this protective effect. To apply this synthetic peptide in the clinical use, a long series of studies is needed to determine the mechanism of action and to identify the range of therapeutic concentrations, which we will examine in further studies. However, at this moment it is too early to provide the assessment of clinical use of this peptide, despite the undoubted fact that PAR1 agonist synthetic peptides are promising basis of novel neuroprotective drugs for disorders related to glutamate excitotoxicity.

Corresponding clarifications, we have included to the text of the manuscript.

4          The paper did not compare the neuroprotective effects of the synthetic peptide analogues to other existing neuroprotective drugs or therapies.

Author response: It was previously shown that thrombin, a canonical PAR1 agonist, causes proinflammatory effects in high concentrations and anti-inflammatory effects in low concentrations, as shown in the work : Gorbacheva L, Pinelis V, Ishiwata S, Strukova S, Reiser G. Activated protein C prevents glutamate- and thrombin-induced activation of nuclear factor-kappaB in cultured hippocampal neurons.Neuroscience. 2010 Feb 17;165(4):1138-46. doi: 10.1016/j.neuroscience.2009.11.027.

  • The study did not provide detailed information on the mechanisms underlying the neuroprotective effects of the synthetic peptide analogues.

Author response:

 One of the neurodegenerative mechanisms is pathological persistent elevation of intracellular calcium. Previously, we have shown in the model of glutamate excitotoxicity that the protective effects of APC are associated with the determination of calcium signaling and the determination of the transcriptional factor NF-kb (Gorbacheva L, Pinelis V, Ishiwata S, Strukova S, Reiser G. Activated protein C prevents glutamate- and thrombin-induced activation of nuclear factor-kappaB in cultured hippocampal neurons. Neuroscience. 2010 Feb 17;165(4):1138-46. doi: 10.1016/j.neuroscience.2009.11.027. Epub 2009 Nov 18. PMID: 19931359). In this work we have tried to analyze the effect of AP9 peptide on calcium signaling, which is one of the factors leading to neuronal death. We demonstrated that against the background of AP9 action neuron death decreases and the level of intracellular calcium stabilizes, so on the basis of this we can conclude that the protective mechanism of AP9 is calcium-dependent. Though, the exact mechanism underlying the neuroprotective effects of AP9 are not discovered yet, it might function similarly to TR47 peptide that represents the N-terminal sequence of PAR1 that exists after cleavage at Arg46. This peptide promotes signaling via β-arrestins, especially β-arrestin-2, and dishevelled-2 (Mosnier LO, Sinha RK, Burnier L, Bouwens EA, Griffin JH. Biased agonism of protease-activated receptor 1 by activated protein C caused by noncanonical cleavage at Arg46. Blood. 2012 Dec 20;120(26):5237-46. doi: 10.1182/blood-2012-08-452169. Epub 2012 Nov 13. PMID: 23149848; PMCID: PMC3537315; Griffin JH, Zlokovic BV, Mosnier LO. Activated protein C: biased for translation. Blood. 2015 May 7;125(19):2898-907. doi: 10.1182/blood-2015-02-355974. Epub 2015 Mar 30. PMID: 25824691; PMCID: PMC4424414).

Corresponding clarifications, we have included to the text of the manuscript.

  • The paper did not discuss the potential limitations or challenges in translating the findings from in vitro experiments to clinical applications.

Author response: Compared to activated protein C, which can cause an immune response and excessive bleeding, peptide analogues - PAR1 agonists - are free of such side effects. In addition, peptide analogues have a wider range of delivery capabilities to target organs and a cheaper synthesis process. In this regard, the search for peptide drugs is more preferable and promising. In comparison to the existing peptide TR47 our peptide AP9 have some benefits as it is shorter and more preferable in terms of synthesis and clinical application.

At this stage, it is too early to talk about the use of this peptide in clinical practice, as further study of its properties and its effect on brain cells is needed to identify potential limitations in use and side effects. More detailed screening of AР9 effects and mechanisms of its action is a broad field for further research.

Corresponding clarifications, we have included to the text of the manuscript.
